# Gas Sensor Based on Lossy Mode Resonances by Means of Thin Graphene Oxide Films Fabricated onto Planar Coverslips

**DOI:** 10.3390/s23031459

**Published:** 2023-01-28

**Authors:** Ignacio Vitoria, Elieser E. Gallego, Sonia Melendi-Espina, Miguel Hernaez, Carlos Ruiz Zamarreño, Ignacio R. Matías

**Affiliations:** 1Electrical, Electronic and Communications Engineering Department, Public University of Navarre, 31006 Pamplona, Spain; 2Institute of Smart Cities, Jeronimo de Ayanz Building, 31006 Pamplona, Spain; 3Telecommunications and Electronic Department, University of Pinar del Río, Pinar del Río 20100, Cuba; 4School of Engineering, University of East Anglia (UEA), Norwich Research Park, Norwich NR4 7TJ, UK

**Keywords:** gas sensor, graphene oxide, Lossy Mode Resonance, optical fiber sensor

## Abstract

The use of planar waveguides has recently shown great success in the field of optical sensors based on the Lossy Mode Resonance (LMR) phenomenon. The properties of Graphene Oxide (GO) have been widely exploited in various sectors of science and technology, with promising results for gas sensing applications. This work combines both, the LMR-based sensing technology on planar waveguides and the use of a GO thin film as a sensitive coating, to monitor ethanol, water, and acetone. Experimental results on the fabrication and performance of the sensor are presented. The obtained results showed a sensitivity of 3.1, 2.0, and 0.6 pm/ppm for ethanol, water, and acetone respectively, with a linearity factor R^2^ > 0.95 in all cases.

## 1. Introduction

The demand for new gas sensors has been increased in the last few years, associated with the development of smart cities or Industry 4.0, which require the collection of significant amounts of data through multiple sensors [1,2]. For example, the monitoring of air quality in cities is crucial, as strict regulations impose limits on the concentrations of several gases either outdoors or indoors [1]. Air quality is a key and very relevant aspect in the development and improvement of smart, highly efficient, and citizen-oriented cities. In fact, the European Commission is currently financing a number of projects along these lines [3]. Gas sensors are also widely used in industry, particularly in the chemical industry, where they are employed to control the production processes, as well as to secure a safe working environment [2]. In addition, several studies have shown that measurements of certain gas concentrations in exhaled breath can aid the diagnoses of different diseases, for example, a high concentration of acetone can be linked to lung cancer [4]. 

Due to the heterogeneous and numerous applications where gas sensors are needed, a wide range of technologies have been developed through the years [5]. Among them, optical sensors have gained popularity because of their excellent features, such as robustness and resistance to harsh atmospheres, along with others [6]. In this context, optical sensors based on the Lossy Mode Resonance phenomenon (LMR) have attracted great attention in the last decade due to their good performance compared to state of the art optical sensors [7,8].

LMRs are generated when a thin film (with particular optical properties that will be later described) is deposited onto a waveguide. In these cases, some modes previously transmitted within the waveguide are guided through the thin film and, consequently, are attenuated. The attenuation occurs at a specific wavelength band, so an absorption peak is obtained in the spectrum, known as LMR resonance. Different guided modes can experience the transition to the thin film, forming resonances of different orders. The LMR resonance has similar characteristics to the more mature Surface Plasmon Resonance [9], although the nature of the phenomenon is different. 

Also, for each resonance, the transverse electric (TE) modes and transverse magnetic (TM) modes can generate different absorption peaks. In contrast SPRs are only generated in the TM mode. The LMR resonance wavelength is highly sensitive to the thickness and optical properties of the thin film, as well as to the surrounding medium’s refractive index. Changes in these parameters produce a shift in the LMR resonance wavelength. The monitoring of this shift can be measured, and it has been exploited in the fabrication of sensors for a wide variety of applications, such as monitoring oil degradation, humidity, pH, organic vapors, biosensing (Igg, D-dimer, gliadin), etc. [8]. 

The majority of LMR based sensors use an optical fiber as the waveguide. Recently, planar waveguides (cover slips) based on the lateral incidence of light, have started to gain popularity [10]. This setup avoids using an optical fiber, which is more brittle, or the Kretschmann configuration (used mainly in SPR-based sensors) [11], which is bulkier and presents difficulties to couple light at angles where LMRs are typically excited (near 90° with the normal of the surface where the thin-film is deposited) [10].

The LMR generating material (thin-film) must fulfill the specification that the real part of the permittivity is positive and higher in magnitude than its own imaginary part and that of the material surrounding the thin film [12]. Metal oxides and polymers usually satisfy the previous conditions. The number of LMR generating materials is wide and heterogeneous in comparison to those enabling the generation of the analogous phenomenon SPR [9] (mainly noble metals, such as gold and silver).

In this regard, graphene oxide (GO) has been found to be an attractive LMR generating coating [13]. Its 2D nature, characterized by a high surface area, enables efficient interactions with the target gas molecules, thereby enabling ultra-sensitive sensors [14]. Additionally, GO can be economically mass produced from graphite [15], making it an attractive option for sensor fabrication. Although there is still a long way to go to exploit its full potential as a sensitive material, it is undoubtedly one of the most promising materials for gas sensing applications [16]. GO has already been employed in gas sensors with promising results [17]. In particular, GO based devices have been successfully used in electronic temperature and humidity sensors [18], as GO-DNA based sensors [19], or as optical fiber LMR-based sensors for the detection of ethanol in water [20], among others. An in-depth study of LMR-based gas sensors, that compares different materials, verifies that GO is a promising candidate for these types of sensors [7].

This work explores the combination of GO thin films deposited onto coverslips, and the LMR sensing phenomenon for the fabrication of gas sensors. GO thin films have been fabricated onto planar waveguides by means of the layer-by-layer (LbL) deposition technique. The sensitivity of the obtained devices to different gases (ethanol, water vapor, and acetone) was studied. To the authors’ knowledge, this is the first time that GO has been included in a planar LMR sensing device for gas detection.

## 2. Materials and Methods

### 2.1. Sensor Fabrication

GO thin films were fabricated onto the waveguides using the LbL technique. Glass microscope coverslips from RS France, 18 mm × 18 mm × 0.15 mm, made from soda lime glass were used as the optical substrates and waveguides [21].

The GO was purchased from Graphenea, S.A. (San Sebastian, Spain) and polyethylenimine (PEI) was obtained from Sigma-Aldrich. LbL fabrication of PEI/GO thin-films is explained in detail elsewhere [22]. A 0.5 mg/mL GO dispersion, and 2 mg/mL PEI solution in DI water, were prepared. The PEI solution was left stirring overnight. GO dispersion was sonicated for 2 h before the deposition to enhance the dispersion of the GO nanosheets. The substrate was cleaned with soap and water and dried with air prior to deposition. The surface of the substrate was activated by immersing it into a solution of KOH (1 M) for 30 min. The LbL deposition started with the immersion of the substrate into the PEI solution for 5 min. Then, the substrate was rinsed with DI water to remove the excess material, and dried in air for 1 min. Afterwards, the substrate was immersed into the GO dispersion for 5 min and subsequently rinsed in water and dried. The described routine formed the first bilayer of PEI/GO. The cycle was repeated multiple times for the deposition of more bilayers to increase the thickness of the film.

The surface homogeneity of the fabricated coating was characterized by means of a scanning electron microscope (NanoSEM 450 FEG from FEI Company, Hillsboro, OR, USA).

### 2.2. Optical Setup

Figure 1 shows the optical transmission configuration for gas measurement. The setup consists of a Taki-MP halogen lamp (from PYROISTECH, Pamplona, Spain) as the excitation source, and a HR4000 VIS spectrometer (from OceanOptics Inc., Largo, FL, USA) for the monitoring of the LMR resonance wavelength shift. Two optical fibers were placed in contact with the facets of the coverslip to couple the light through the waveguide.

The sensor was tested with different concentrations of gases at a constant flow rate during measurements (200 mL/min). The chamber consisted of a sealed cylindrical cavity made of stainless steel, with an inner diameter of 8.4 cm and a height of 1.5 cm. A gas flow controller (EL-Mass flow Meters from Bronkhost®, Ruurlo Netherland) regulated the flow of the carrier gas (nitrogen N_2_). N_2_ gas was mixed with the gases under study (ethanol, water, and acetone) that were evaporated using a Controller Evaporated Mixer (CEM) from Bronkhost®. The conduit between the chamber and the panel had a total length of 55 cm with an inner diameter of 5 mm. Due to the total system volume of the chamber (94 cm^3^ including the conduit volume) and the selected flow rate, changes in the gas concentrations were not instantaneous. In fact, approximately 28 s were required to fully replace the gas inside the chamber.

The response and recovery times were estimated as the required time (when a new gas concentration was set) to go from 10% to 90% of the maximum wavelength shift value and vice versa.

## 3. Results and Discussion

Previous studies have shown the possibility of generating LMRs in optical fibers with PEI/GO thin films [20], however the generation of an LMR onto a planar waveguide, in this case a coverslip, has not been previously studied. In the present work, a resonance near 760 nm (resonance wavelength) was obtained after depositing 35 bilayers of PEI/GO. The number of bilayers exhibits a proportional relationship with the resonance wavelength (an increase in the number of bilayers results in a shift towards longer wavelengths [8]). In this study, 35 bilayers were selected to place the resonance within the region of the spectrum under examination (visible region). As can be seen in Figure 2b, the sensitive thin film is very uniform and homogeneous. In the SEM micrograph (Figure 2c) it is possible to observe some wrinkles due to the folding of the GO nanosheets.

The resonance generated by this sensitive coating is wide compared to that generated by other materials [23]. Particularly, the width of the resonance between two points that are 0.1 dB higher than the center of the resonance is 501 nm (see Figure 2a). This characteristic can be attributed to the structure of the GO sensing film, as the presence of a polymeric matrix within GO nanosheets results in a rough material. This characteristic was also observed with other coatings made of polymers and nanoparticles [24,25]. To consistently obtain the wavelength positions of the peaks at maximum absorbance, an algorithm that models absorbance spectral data with parabolic curves was created. Additionally, a sliding window algorithm was employed to reduce the noise.

The fabricated device was tested in a gas chamber with a mixture of N_2_ and the previously mentioned evaporated gases at a constant flow of 200 mL/min. First, three cycles, alternating a concentration of 0 and 50 mg/h of the selected gas, were executed to check the repeatability of the measurements. Next, a few more cycles with different concentrations were performed to obtain the sensitivity of the device under testing. Although the gas controllers were setup in mg/h, the concentrations of the gases are represented using more standard units (ppm in volume).

Wavelength resonance shifts associated with increasing concentrations of ethanol, acetone, and water vapor are shown in Figure 3a–c, respectively. It can be seen from Figure 3 that the LMR shifts to higher wavelengths in all cases when increasing the gas concentration. The variations and noise presented when the LMR is stabilized at the different gas concentrations can be attributed to slight inaccuracies of the CEM that fix the concentrations of the gases. The shift of the LMR for ethanol was measured with concentrations between 600 and 1820 ppm (Figure 3a). Lower ethanol concentrations could not be reached due to limitations in the gas flow controllers. The average ethanol response and recovery times were 123 s and 263 s, respectively.

The acetone gas concentration was studied in the range between 3200 and 6390 ppm (Figure 3b), obtaining an average response and recovery time of 187 s and 469 s, respectively. The response to water vapor is shown in Figure 3c. In this case, the measured response and recovery times were 49 s and 696 s, respectively. This fast response time can be attributed to the high capillary-like pressure in GO laminates [26].

The response of the device showed good repeatability for ethanol, acetone, and water vapor for several cycles. The resonance wavelength shift versus gas concentration is represented in Figure 4 to obtain the sensitivity of the sensor to the three selected gases. In the case of ethanol, the increase of the wavelength is linear to the concentration of the gas in the measured range (R^2^ = 0.989), with a sensitivity of 4.826 × 10^−3^ nm/ppm. The error bars represent the standard deviation of the measurements associated with inaccuracies of the CEM. The sensitivity of the device to acetone was remarkably inferior to that of ethanol (26.2% lower), with a sensitivity of 1.266 × 10^−3^ nm/ppm and an R^2^ value of 0.951. The sensitivity of the device to water vapor was 1.367 × 10^−3^ nm/ppm, with a linearity factor of R^2^ = 0.989 in the studied range. In this case, the CEM evaporated water steadily, so the standard deviation is lower than that obtained with ethanol or acetone. 

The combination of the linear response of the sensor and the high-resolution of the spectrometer (0.25 nm) allows the detection of small concentrations of the gases under study. Specifically, in this work, the smallest variations that can be observed for ethanol, acetone, and water vapor are 52, 197, and 183 ppm, respectively. As shown in Figure 4, the trendlines do not intersect the origin of the axes, revealing a non-linear response of the device at lower gas concentrations (towards 0 ppm), which could lead to the detection of much smaller amounts using a more sophisticated equipment and more accurate gas controllers. In particular, ethanol detection shows promising results, with the largest resonance shift (23 nm) at 600 ppm. 

The obtained parameters for each gas are summarized in Table 1. The sensitivity of the device to ethanol is ~3.5 times higher than the sensitivity to acetone and water vapor in the measured ranges. Moreover, the response time of the fabricated device to ethanol has the shortest recovery time. However, the best response time was obtained in the presence of water vapor, being 2 times shorter in comparison to the other two gases under study. This fact could be ascribed to the polarity of the gas molecules. In particular, the relative polarity of the selected gases is 0.654 for ethanol, 0.355 for acetone, and it is especially high for water, 0.998 (normalized from measurements of solvent shifts of absorption spectra [27]). Therefore, the numerous functional groups present on the surface of GO particularly facilitate the intercalation of water molecules.

The literature regarding LMR-based gas sensors utilizing GO is scarce. While there exist studies that investigate GO-based optical gas sensors, these often employ different phenomena, gas chambers, or waveguides, thus hindering direct comparison, particularly with regard to sensitivity. Table 2 presents an overview of the primary parameters of these investigations. It is worth noting that the detection limits of the studied sensor has not been fully established using the current gas setup, and additional research is needed to achieve this goal and to optimize the sensor’s performance

## 4. Conclusions

An LMR based gas sensor was developed using GO as the sensing material and a coverslip as the waveguide. The probe was tested against ethanol, acetone, and water vapor in a gas chamber with a constant flow and utilizing an optical transmission setup. The device showed good repeatability and sensitivity to the three selected gases, especially in the case of ethanol (4.826 × 10^−3^ nm/ppm at room temperature). Sensitivities for water vapor and acetone were 1.367 × 10^−3^ and 1.266 × 10^−3^ nm/ppm, respectively. The device revealed a high linearity in the studied range for the gases under testing. This work shows a promising approach for the fabrication of gas sensors combining the utilization of GO and LMR-based sensing technology, as a preliminary step to attaining highly sensitive devices, and the advancement in the progress towards the commercialization of a lab-on-chip type sensor.

## Figures and Tables

**Figure 1 sensors-23-01459-f001:**
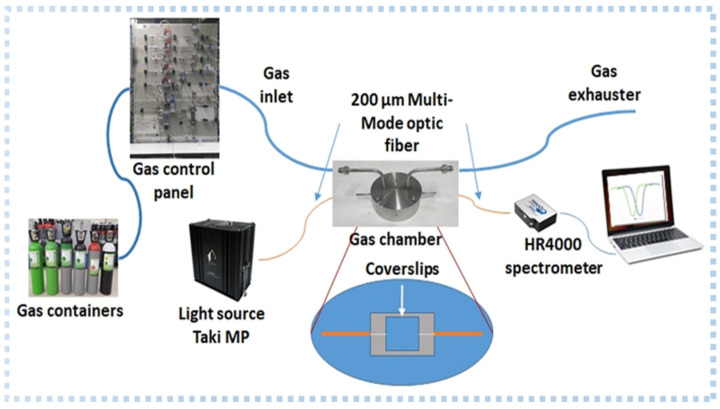
Gas measuring setup.

**Figure 2 sensors-23-01459-f002:**
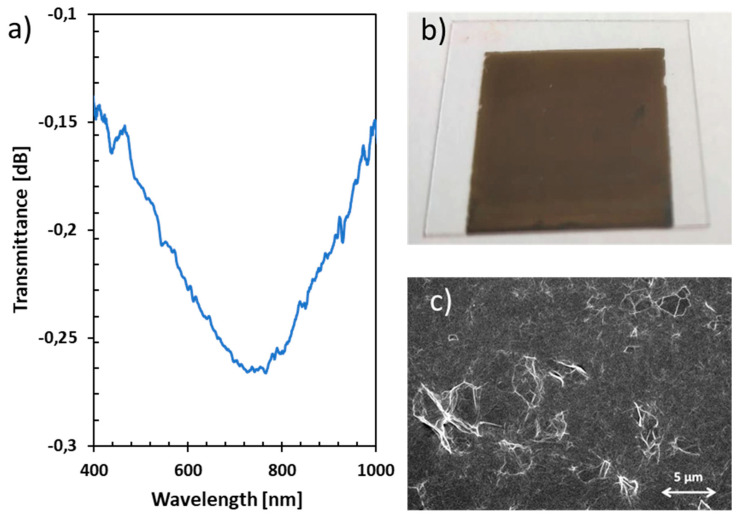
(**a**) Transmittance spectra of the LMR with PEI/GO of 35 bilayers, (**b**) Coverslip coated with PEI/GO, (**c**) SEM micrograph of the PEI/GO thin film.

**Figure 3 sensors-23-01459-f003:**
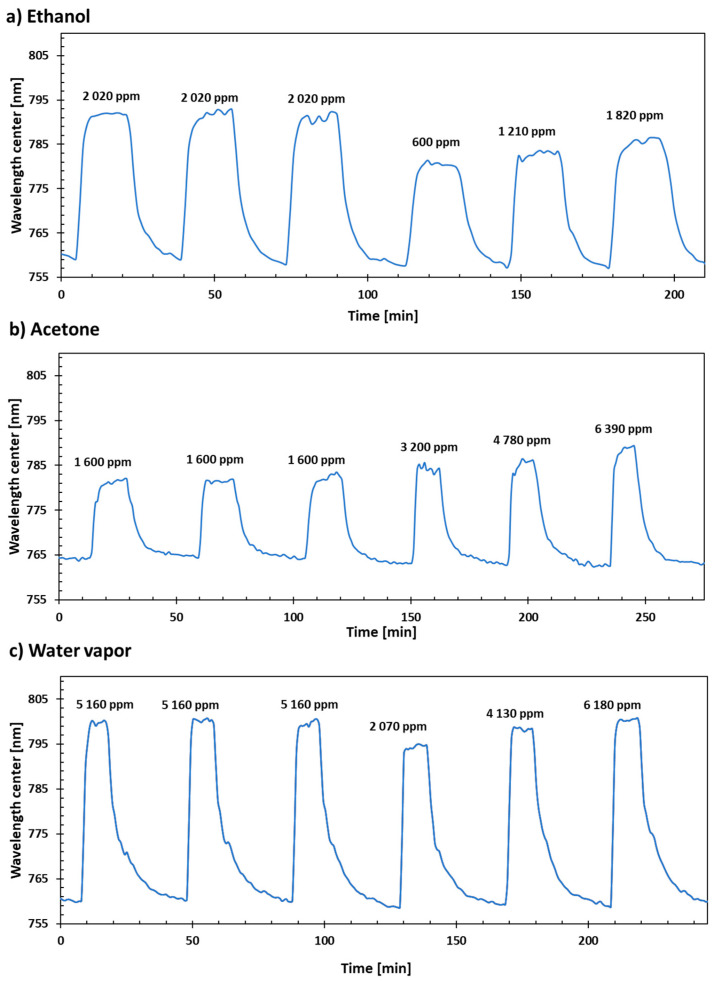
Dynamic response of the fabricated device to different concentrations of (**a**) ethanol, (**b**) acetone, and (**c**) water vapor.

**Figure 4 sensors-23-01459-f004:**
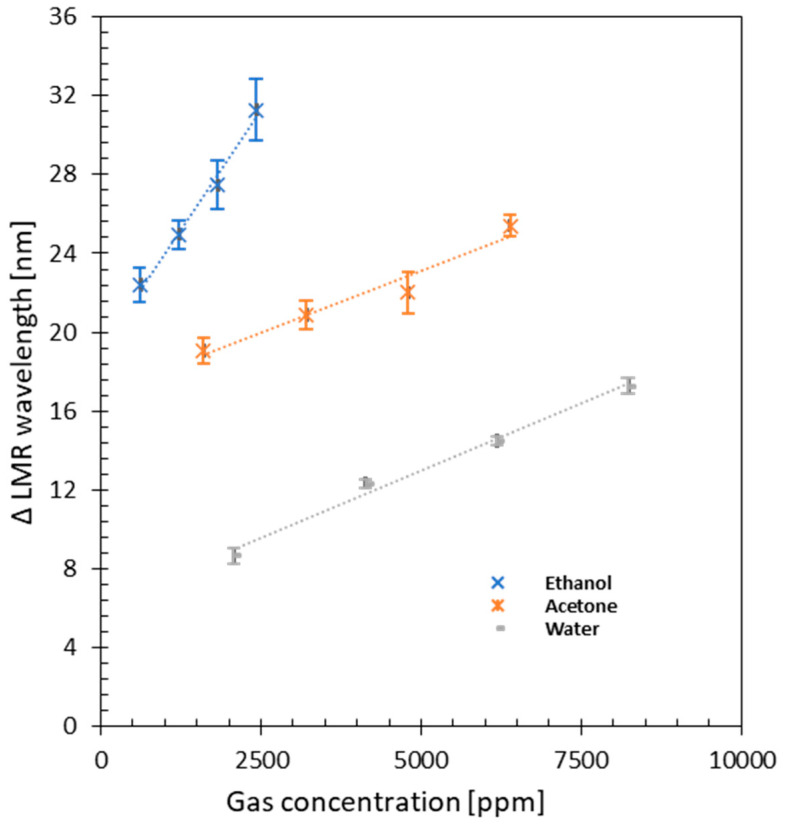
Sensitivity of the sensor, shift of the resonance, to the different gases: ethanol, acetone, and water.

**Table 1 sensors-23-01459-t001:** Summarized response parameters of the sensor to different gases.

Gas	[GAS]_ppm_	Sensitivity (nm/ppm)	Response Time (s)	Recovery Time (s)
Ethanol C_2_H_5_OH	600–2020	4.826 × 10^−3^	123	263
Acetone C_3_H_6_O	1600–6390	1.266 × 10^−3^	187	469
Water H_2_O	2070–6180	1.367 × 10^−3^	49	696

**Table 2 sensors-23-01459-t002:** Summary of parameters for sensors with GO in the literature.

Gas	[GAS]_ppm_	Detection Limit	Waveguide	Phenomenon	Reference
(1) Ethanol (2) Acetone (3) Water	(1) 600–2020 (2) 1600–6390 (3) 2070–6180	-	Coverslip	LMR	Present work
Ethanol Ammonia Methanol	0–500	100 ppm *	Plastic optical fiber	Evanescent field	[28]
Ethanol	0–80	16 ppm *	Tapered polariztion-maintining fiber	Interferometer	[29]
(1) Ethanol (2) Acetone Others	(1) 164–823 (2) 130–650	(1) 164 ppm * (2) 130 ppm *	Planar waveguide	Plasmonic nanospots	[30]
Ethanol Methanol	0–500	100 ppm *	Plastic optical fiber	Evanescent field	[31]
Humidity (Water)	-	30% Relative humidity	Multimode optical fiber	LMR	[32]
Humidity (Water)	-	20% Relative humidity	Type D single mode fiber	LMR	[33]

* Data not specified. The values are the minimum concentration found in the articles.

## Data Availability

Data underlying the results presented in this paper are not publicly available at this time but may be obtained from the authors upon reasonable request.

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
