# Peer review of "Gas Sensor Based on Lossy Mode Resonances by Means of Thin Graphene Oxide Films Fabricated onto Planar Coverslips"

_sensors, 2023, doi:10.3390/s23031459_

Round 1

Reviewer 1 Report

This design combines GO with LMR for the first time, and proposes a new sensor design concept. It has realized the sensing measurement of ethanol, water and acetone, which has certain innovation. At the same time, the experimental verification also has some significance in practical application. However, there are still several areas that can be improved:

1. The calculation of sensing accuracy has not been carried out. It is recommended to add error analysis and other data to further verify the practical application value of the design.

2. The paper only describes its own design work and does not compare with the known research. It is recommended to add literature comparison to highlight whether the design has better sensing performance.

3. The reasons for the use of graphene oxide are not clear. Why the use of this material film? It is recommended to supplement the advantages of this material with other materials or supplement the data of other materials for comparison to improve the rigor of the paper.

Reviewer 2 Report

COMMENTS FOR AUTHOR

The paper under consideration entitled “Gas sensor Based on Lossy Mode Resonances by means of the Graphene Oxide films fabricated into Planner Coverslips” deals with the gas sensor based on lossy mode resonance technology made upon the planner waveguide coated with graphene oxide. The sensor fabricated finds its application as ethanol, water vapor and acetone sensing. The topic and the device are relevant and interesting for the readers of this journal. Despite its scientific contents, the paper needs improvements before its publication.

General concept comments

1.      The paper under consideration has not clearly mentioned the gap of the study. The transition from optical sensor to waveguide sensor has to be briefly explained. The recent references are less cited.

2.      Lossy Mode resonances are to be compared with surface plasmon resonances and others if any. I suggest authors to refer to doi.org/10.1007/s13538-022-01064-0, 10.24425/opelre.2021.139601 and like others, if you wish to compare, to add some text on this regard.

 Specific comments 

In the result and discussion section, Table 1 summarizes the response parameters. Add detection limit as one more parameter. Also, validate your results by comparing with other results and show the position of your results.

Reviewer 3 Report

The authors presented a graphene based gas sensor however there are some points that should be addressed, mainly, concerned with the used parameters. It appears that they implemented the gas sensor without any design rule and made the characterization and just reported it.

What was the criteria to choose the usedcwavelength? Why not IR ou THz?

How the wavelength impacts the design of the sensor? in terms of sensitivity or the geometrical parameters. 

Why did they consider 35 bilayers? How the response is affected by changing this number?

The authors did not compare their results with any other of the literature to show what has been improved here. Is the present sensor more sensitive? It is a more linear behavior? Does it exhibit a faster response?

Is the time response of the presented sensor good enough for practical applications? If so, for which ones?

Did the inclusion of graphene improve the response of the sensor? How much was this improvement?

Round 2

Reviewer 1 Report

After revision, I think this is a more detailed research paper. The content of the demonstration is rich and comprehensive, which provides a reasonable supplement and explanation to the problem of interpretation. With high scientific rigor and detailed experimental data, it can be used by journals.

Reviewer 3 Report

Authors have addressed all the issues pointed out during the review process.